# Consumer Intention to Participate in E-Waste Collection Programs: A Study of Smartphone Waste in Indonesia

**Diana Puspita Sari** [1,2,*], **Nur Aini Masruroh** [1] **and Anna Maria Sri Asih** [1]

1   Department of Mechanical and Industrial Engineering, Universitas Gadjah Mada,
    Yogyakarta 55281, Indonesia; aini@ugm.ac.id (N.A.M.); amsriasih@ugm.ac.id (A.M.S.A.)
2   Department of Industrial Engineering, Diponegoro University, Semarang 50275, Indonesia
*   Correspondence: dianapuspitasari@lecturer.undip.ac.id

**Abstract:** Indonesia is a developing country with a low-level e-waste management system based on a limited number of informal initiatives. E-waste requires proper management procedures, which involve the design of a reverse logistics management network. Consumers play a critical role in such a network, because the network runs when they willingly participate as suppliers of waste. This paper applies the Theory of Planned Behavior framework and extends it using reverse logistics drivers, the value belief norm theory, and facility accessibility to explain consumer intention to participate in e-waste collection programs. A survey was conducted on smartphone users in Indonesia, with a total of 324 valid questionnaires. The results showed that government drivers, facility accessibility, and personal attitudes significantly influence consumer intentions. Environmental concern has a positive influence on consumer intentions through the variables of the Theory of Planned Behavior and perceived behavioral control through government drivers. This study shows the need for integration, because the variables reinforce each other. However, neither economic drivers nor subjective norms significantly influence consumer intentions. This finding distinguishes Indonesia from other countries, especially developed countries, in that e-waste collection programs have not become part of the culture in Indonesia. For this reason, Indonesia needs regulations, as the most influential variable, to regulate the implementation of such a program.

**Keywords:** Theory of Planned Behavior; reverse logistics drivers; value belief norm theory; facility accessibility; consumer intentions; e-waste

## 1. Introduction

The rapid development of technology and the economy in the last few decades has led to a more diverse choice of electronic products at an affordable price, thus increasing their consumption [1]. When people's consumption of electronic products increases, the potential for e-waste also rises. Electronic waste (e-waste) management has become one of the most challenging problems worldwide, especially in developing countries [2]. This is because it is one of the fastest-growing waste categories globally, reaching an annual growth rate of 3–5% [3]. Developing countries, such as Indonesia, struggle to handle large volumes of e-waste [4]. The Global E-waste Monitor 2017—Quantities, Flows, and Resources—states that the e-waste produced by the Indonesian population is estimated to have been 1.274 million tons or an average of 4.9 kg per capita in 2016. These data position Indonesia as the 9th producer of e-waste worldwide.

Smartphones are among the electronic products with the largest proportion of e-waste. Due to the coronavirus pandemic, there was a surge in smartphone usage worldwide. During this pandemic, people worked and studied online, leading to electronic communication devices, especially smartphones. The usage proportions are 70%, 40%, and 32% for smartphones, laptops, and personal computers, respectively [5]. However, the high potential for this flow has not been matched by good waste management [6]. In Indonesia, no regulations control the system for collecting and transporting e-waste until the final

processing. The United Nations University ranks the waste management system in Indonesia at the lowest level, because it is still limited to informal initiatives. This management pattern differentiates Indonesia from developed countries [7].

Improper waste handling can cause environmental pollution. In general, e-waste is one of the most ozone-depleting substances, with high environmental impacts [8]. For this reason, it is necessary to design an e-waste management system to aid in the minimization of the impact on the environment and increase the economic value of waste. While e-waste is classified as hazardous waste, it has a significant value recovery potential [3]. Due to the negative impacts on humans and the environment caused by the mishandling of e-waste, proper e-waste management procedures, including a reverse logistics (RL) management network, should be in place. RL reuses used products in order to reduce waste and improve industrial environmental performance [9]. Goods and information flow in the opposite direction from forward logistics activities, which support a product and return goods for recycling, manufacturing, reuse, or destruction for disposal [10]. Collection is the starting point of the RL process, with the consumer as the supplier. A sorting process is then carried out, and the appropriate further handling process is determined, which may include repairing, reusing, remanufacturing, recycling, or disposing of the waste [11]. RL drivers include the economy, laws and regulations, and corporate responsibility [12,13]. Srivastava [14] and Mwanza et al. [15] also stated that the economy and laws and regulations are the driving factors of the implementation of RL, but the third factor is not the company's responsibility, but rather environmental concern. This contradicts Chileshe et al. [16] and Chiou et al. [17], who stated that economic, environmental, and social factors drive the success of RL in practice. Encouragement from consumers who are aware of environmental friendliness and government support are two external factors that play an essential role in encouraging RL's creation [18]. Agrawal et al. [19] stated that the most important driving factors for RL in practice are 12 supporting factors, including government regulations; economic factors; environmental concern; top management awareness; resource management; management information systems; requirements and contract terms; direct and indirect taxes; forward and backward supply chain integration, joint consortia, process capabilities, and skilled workers; and consumer awareness and social acceptance. According to Mangla et al. [20], RL practice is driven by regulatory, human and organizational, economic, and strategic factors.

The collection and management of waste is an effort to create a sustainable supply chain. For this reason, an e-waste RL management network whose implementation is supervised and regulated by the government is needed. Consumers play a critical role in the successful implementation of waste management networks. Whether the network operates or not is contingent upon consumers' willingness to participate as suppliers of waste. The Theory of Planned Behavior (TPB) model is often used to analyze consumer behavior. Research related to consumer behavior in relation to environmental issues using TPB has been conducted. TPB is used to analyze the end-of-life (EoL) handling of recycling [21–26]. Santoso and Farizal [27] measured community participation in household waste management, while Arifani and Haryanto [28], Chen and Deng [29], Yadav and Pathak [30], Liobikiene et al. [31], and Mancha and Yoder [32] used TPB to analyze green behavior. Pramod [33] and Tolkes and Butzmann [34] analyzed pro-environmental behavior in general. Apart from TPB, the Norm Activation Model (NAM) is often used. Werff et al. [35] and Zhang et al. [36] used NAM to analyze EoL behavior. Similarly, Moller et al. [37] used NAM to analyze behavior in the transportation sector, while in [38], NAM is used to determine behavior in relation to clothing consumption and reduce it. Pro-environmental behavior has been determined on the basis of NAM [39–42]. Another model commonly used is the Value Belief Norm Theory (VBNT). It was used by Dursun et al. [43], for example, to analyze responsible post-consumption behavior in the form of recycling and reuse in transport. Additionally, Liobikiene and Poškus [44], Ghazali et al. [45], Han et al. [46], and Wynveen et al. [47] used it to analyze pro-environmental consumer behavior.

TPB is one of the most frequently used theoretical models to explain the relationship between intention and behavior. According to this model, behavior is determined by personal intention, which is, in turn, based on attitude, subjective norms, and perceived behavioral control. However, TPB has often been criticized for contradicting the weak correlation of attitudes in social psychological studies [48]. In pro-environmental behavior, individual attitudes reflect personal norms regarding environmental protection. In the context of pro-environmental behavior, individual attitudes reflect personal norms towards environmental protection. However, the influence of personal norms on behavior is simplified in the TPB model, so it is necessary to include indicators of personal norms in the further development of the TPB model [36]. NAM highlights the importance of personal norms for predicting individual behavior. Several studies have expanded the TPB model by integrating NAM [36,49–51]. VBNT combines the components of values and norms from NAM and the New Environmental Paradigm (NEP), including common beliefs and concerns about the environment and the need for action to address problems [44].

The purpose of this study is to analyze consumer intention to participate in e-waste collection programs using the TPB model, which is extended by employing RL drivers and the values of NEP from VNBT and accessibility facility. Consumers bring their used smartphones to collection centers provided by the government or organizations. They participate in e-waste collection programs by bringing their used smartphones to collection centers. This study explores the factors that influence consumer intention to participate in e-waste collection programs and provides suggestions for improvements. The main objective of the waste management process is to protect the environment. TPB is disadvantageous because it is not explicitly structured to recognize moral support for environmental action, and it does not offer psychological constructs to support moral pro-environmental behavior [48]. By integrating TPB and VBNT, the TPB model is refined by emphasizing morality, which combines components of values and norms with a new environmental paradigm. TPB is based on self-interest and considerations following rational choices (factors that reflect perceived attitudes and possibilities), while VBNT focuses on moral values and norms [52]. The values of the new environmental paradigm from the VBNT include biospheric, altruistic, and egoistic values [53]. The integration of TPB and VBNT was used to assess the effect of altruistic values on purchasing decisions in relation to energy-efficient equipment [54] and investigate the intention to purchase organic food [55] in Vietnam, Southeast Asia. Waste collection activities are highly dependent on the participation of consumers as suppliers. RL drivers encourage companies to carry out reverse logistics activities, which promote consumer participation [56]. Previous studies showed that there are three main supporting factors that have been agreed upon: economic reasons, government regulations, and concern for the environment [14,15,18,19]. This study includes facility accessibility, because consumers need easy access to collection centers for participating in e-waste collection programs. According to Zhang et al. [36] and Zhang et al. [48], facility accessibility significantly affects waste sorting behavior and individual intentions to recycle waste.

## 2. Conceptual Models and Hypotheses

Consumers play a critical role as suppliers in reverse logistics activities. An action taken by a consumer has an impact on reverse logistics activities [57]. Specifically, consumer behavior affects the effectiveness of reverse logistics activities, including reusing, recycling, and remanufacturing waste, as well as participation in waste management [56].

The intention to behave in a certain way is a stage where a person wishes to perform certain actions or behaviors. It is the direct antecedent of behavior [58] and is accepted as its best predictor because it represents a person's motivation to exert effort in order to act or behave in a certain way [59,60].

Reverse logistics drivers are factors that influence consumer intention to participate in e-waste collection programs. When these factors encourage companies to carry out reverse logistics activities, they affect consumer participation behavior [56]. Reverse logis-

tics drivers include government drivers and economic drivers, as well as environmental concern [14,15,18,19].

According to Verma et al. [61], environmental concern is influenced by the new environmental paradigm (NEP) from the VBNT, including biospheric, altruistic, and egoistic values. VBNT includes values and personal norms that are not considered in the TPB [48]. In this study, personal attitudes, specifically the integration of personal norms and attitudes, are used [36]. Environmental concern can have a direct effect on behavioral intentions and an indirect effect by passing the TPB variables first, which include attitudes, subjective norms, and perceived behavioral control [62,63]. According to Verma et al. [61], Prakash et al. [64], and Setyawan et al. [65], only one TPB variable is passed, namely, the attitude before the intention variable. Facility accessibility, besides having a significant positive effect on consumer intentions, also significantly affects perceived behavioral control, which is one of the TPB variables [36]. Perceived behavioral control indirectly affects consumers' intention to participate through government drivers as a mediator variable [56,66,67].

Figure 1 shows the proposed framework structure, which uses TPB as a basic model (backbone) and extends it using the NEP values from the VBNT, reverse logistics drivers, and facility accessibility. A total of 15 hypotheses were taken from 11 variables/factors/constructs, including personal attitudes, subjective norms, perceived behavioral control, environmental concern, government drivers, economic drivers, facility accessibility, biospheric values, altruistic values, egoistic values, and consumer intention to participate in e-waste collection programs. The following describes the factors that influence consumer intention to participate in e-waste collection programs.

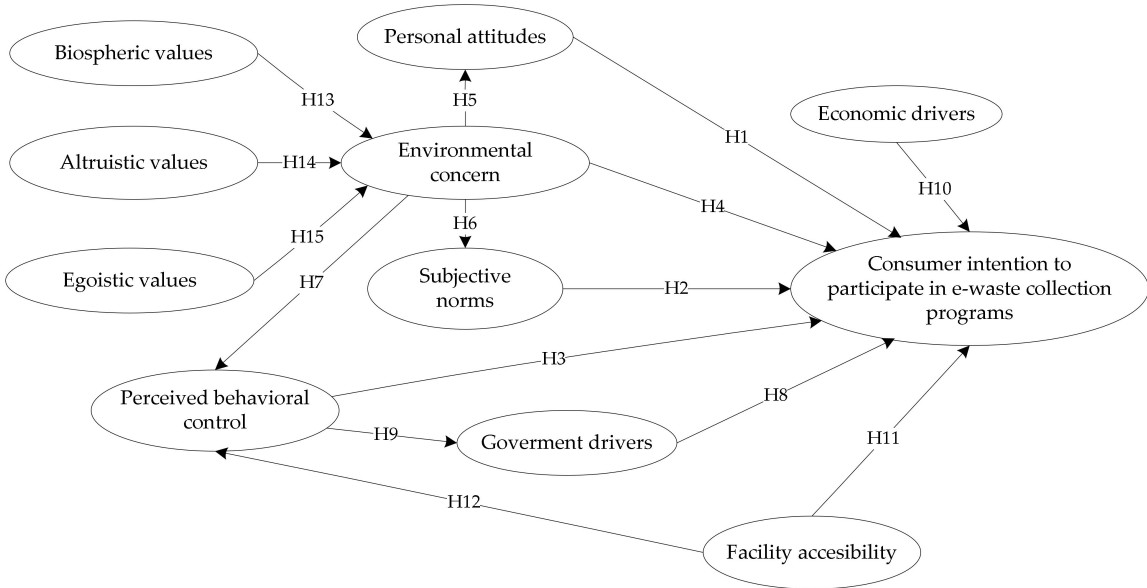

**Figure 1.** Conceptual model and research hypotheses presented in the study.

## 2.1. Theory of Planned Behavior Model

The Theory of Planned Behavior (TPB), introduced by Icek Ajzen in 1985, extends the Theory of Reasoned Action (TRA) by adding perceived behavioral control. TPB is used to predict intentions, which constitute the central dependent variable of TPB [68]. Intentions were the best predictor available for human behavior and are at the core of the TPB framework [30,31]. They indicate a person's readiness to behave in a certain way [68]. This means a conscious plan of action, which specifically requires behaviors and motivations [69]. Several studies have only predicted intentions, assuming that intentions are a good predictor of behavior and fully mediate the impact of attitudes, subjective norms, and perceived behavioral control [31,70,71]. TPB has been used to predict recycling behavior [21–26], green behavior [28–32], green product purchasing [62,65], and energy

conservation [72]. Several studies conducted in developing countries in the Southeast Asia region have used the TPB to analyze intentions to adopt climate change mitigation behavior, green purchase behavior, and recycling behavior [73–75].

### 2.1.1. Personal Norms and Attitudes

An attitude towards a behavior refers to the extent to which a person has a favorable or unfavorable evaluation of a certain behavior [68]. This is an interaction in memory between a given object and a summary of its evaluation [62]. An attitude is a psychological emotion, which is directed through consumer evaluation. A behavior tends to be more positive if the psychological emotion is positive [76]. Personal norms are intrinsic factors that reflect individual subjective desires and relate to self-expectations and individual attitudes stemming from related norms and values regarding certain behaviors [36]. Collecting post-use products is an altruistic behavior, where people need to sacrifice their time and convenience for the benefit of recycling practices. A person's intention to engage in altruistic behavior in collecting waste cannot be cultivated without the intrinsic motivation of personal norms. Previous studies show that the behavioral intention to buy green products and engage in waste recycling depends on attitudes [21,23,29,30,62,63]. Personal norms significantly influence the behavioral intention to reduce clothing consumption and purchase green products [38,41,47]. According to Liu et al. [49] and Zhang et al. [36], the integration between attitudes and personal norms has a significant positive effect on reducing car travel intentions and increasing waste-sorting intentions. By combining TPB and VBNT, personal norms and attitudes, referred to as personal attitudes, are likely to increase waste collection behavior, which can be explained through behavioral intentions. Therefore, the following hypothesis was formulated.

**Hypothesis 1 (H1).** *Personal attitudes are positively related to consumer intention to participate in e-waste collection programs.*

### 2.1.2. Subjective Norms

In the TPB model, the second antecedent of a behavioral intention is a subjective norm, which is a felt social pressure to do or not do something [62,68], highlighting the influence of other people who are close/important to the actors, such as family, close friends, relatives, and colleagues. Consumers with positive subjective norms often have positive behavioral intentions [70]. Subjective norms promote the intention to recycle agricultural waste in China [77] and encourage the behavioral intention to recycle plastic waste in developing countries [78]. For this reason, the following hypothesis was formulated.

**Hypothesis 2 (H2).** *Subjective norms are positively related to consumer intention to participate in e-waste collection programs.*

### 2.1.3. Perceived Behavioral Control

The third TPB antecedent is perceived behavioral control, which is the ease or difficulty that a person feels in performing certain behaviors [62]. Perceived behavioral control and motivation determine behavior [79]. According to Coskun et al. [80], it is the most important predictor of the intention to reduce food waste among all TPB independent variables. There was a positive relationship between perceived behavioral control and waste sorting behavior associated with household kitchens in Beijing, China [81]. Similarly, Nguyen et al. [75] proved that there is a significant relationship between perceived behavioral control and the intention to separate household waste in developed countries using case studies in Vietnam. Nadlifatin et al. [66] and Persada et al. [67] said that perceived government regulations/support regarding environmental factors positively affect environmental behavior. Perceived government regulations have a strong indirect effect on takeback intentions through government drivers [56]. For this reason, the following hypotheses were formulated.

**Hypothesis 3 (H3).** *Perceived behavioral control is positively related to consumer intention to participate in e-waste collection programs.*

**Hypothesis 4 (H4).** *Perceived behavioral control is positively related to government drivers.*

*2.2. Reverse Logistics Drivers*

Previous studies have shown that there are three driving factors of RL for the success of reverse logistics. The first driving factor is environmental concern [14–19], followed by regulation/government [12,13,18–20] and the economy [12,13,16,17,19,20].

### 2.2.1. Environmental Concern

Environmental concern is the extent to which people are aware of environmental problems and their willingness to solve them [62] and contribute personally to their solutions [82]. These problems have long been considered important predictors of ecological decisions [61,63,64,83–85]. Therefore, individual concern is a valuable driver of environmentally conscious behavior, which ranges from green purchasing behavior and recycling to energy conservation [43,62–65,72]. Previous studies have proven that environmental concern affects consumer attitudes [55,61,64,65,86]. According to Maichum et al. [62] and Paul et al. [63], environmental concern also affects subjective norms and perceived behavioral control, leading to purchasing intentions. Therefore, the following hypotheses were formulated.

**Hypothesis 5 (H5).** *Environmental concern significantly and positively influences consumer intention to participate in e-waste collection programs.*

**Hypothesis 6 (H6).** *Environmental concern has a positive and significant effect on personal attitudes.*

**Hypothesis 7 (H7).** *Environmental concern has a positive and significant effect on subjective norms.*

**Hypothesis 8 (H8).** *Environmental concern has a positive and significant effect on perceived behavioral control.*

### 2.2.2. Government Drivers (Regulation)

Government drivers refer to government authorities' regulations or actions that relate to the minimization of the effects of end-of-life products on the environment [19]. E-waste policies and laws play an essential role in establishing the principles and responsibilities of stakeholders [2]. Legislation is one of the determinants of RL's success, because the law is the driving force of the system [87]. In many developed countries, conceptual laws work when the government forces companies to address the problem of proper disposal and restore the value of end-of-life products [88]. The influence of legislation regarding RL products has been noticed in the electronics industry. Environmental problems caused by waste disposal practices can trigger a country to issue rules and regulations governing waste practices [89]. For instance, Lambert et al. [90] described the European Union (EU) directive on waste electrical and electronic equipment (WEEE). Laws and regulations compel manufacturers to set up product recovery and safe disposal systems in several industries [15]. Compliance with regulations is the most vital element for motivating recycling behavior [25]. Therefore, the following hypothesis was formulated.

**Hypothesis 9 (H9).** *Government drivers have a significant and positive effect on consumer intention to participate in e-waste collection programs.*

### 2.2.3. Economic Drivers

Economic benefits contribute significantly to product recovery [15]. Economic reasons are the motivation for implementing RL, because incentives stimulate citizens to sort and place waste properly in designated collection points [91]. According to Akdoğan and Coşkun [13], direct and indirect benefits in all recovery measures are related to economic

benefits. Economic measures in the form of incentives and penalties are needed to orient motivations in the desired direction [87]. The community ranks economic incentives as the main driver of waste recycling [25]. Therefore, the following hypothesis was formulated.

**Hypothesis 10 (H10).** *Economic drivers have a significant and positive effect on consumer intention to participate in e-waste collection programs.*

### 2.3. Facility Accessibility

Accessibility reflects comfort and convenience, though it is different from perceived behavioral control [36]. It has a very similar connotation to availability [48]. According to the TPB model [68], perceived behavioral control, which refers to an individual's perception of their ability to perform certain behaviors, is one of the main predictors of behavioral intention. Accessibility is an objective condition for implementing a certain behavior. Facility accessibility sometimes affects perceived behavioral control, for example, when a person is aware of it [36]. The facility's location plays a vital role in building a financially sustainable waste management system [92]. Perceived distance is an essential factor influencing recycling behavior [93]. Shorter distances and easier access to sorting facilities save time in recycling waste and encourage personal behaviors [94]. Having adequate space for the temporary storage of sorted materials can increase waste-sorting rates and recycling behavior [95]. Roadside facilities for sorting waste, including a clear classification logo on trash and a sufficient capacity at each waste sorting place, are aspects of accessibility related to waste separation behavior [36]. The following hypotheses were proposed.

**Hypothesis 11 (H11).** *Facility accessibility is positively related to consumer intention to participate in e-waste collection programs.*

**Hypothesis 12 (H12).** *Facility accessibility is positively related to individual perceived behavioral control.*

### 2.4. Value Belief Norm Theory (VBNT)

Value Belief Norm Theory (VBNT) focuses on moral values and norms [52]. It combines NAM's values and norms and the new environmental paradigm (NEP) [44]. The new environmental paradigm values of VBNT include biospheric, altruistic, and egoistic values [53]. One's value orientation—especially biospheric, altruistic, and egoistic orientations—determines one's pro-environmental behavior [45,46,96]. People with strong biospheric values focus on consequences for nature and the environment, while those with strong altruistic values focus more on the consequences for other people. People with strong egoistic values will respect and focus on the consequences for their own resources [97].

#### 2.4.1. Biospheric Values

Biospheric values constitute a person's pro-environmental behavior, which is based on the perceived costs and benefits for an entire ecosystem [96]. Individuals with strong biospheric values have a primary concern for nature (environment) and make decisions based on the costs and benefits to an ecosystem [98]. The concern for the biosphere, environment, and ecosystem involves values that emphasize the environment and the biosphere [45]. Biospheric values emphasize the quality of the environment, rather than the benefits it provides to humans [99]. Previous studies have shown that biospheric values could be a strong predictor of the new environment paradigm, ecological worldview, behavioral intentions, attitudes towards pro-environmental behavior, and environmental norms [46,52,96,100]. The biospheric value orientation is consistently and significantly related to individual environmental behavior [101,102]. Additionally, Hiratsuka et al. [97] and Ghazali et al. [45] argued that consumers with a strong biospheric value orientation would have a high awareness of the environmental consequences. The biospheric value orientation significantly influences environmental concern [61]. When associated with

one of the RL drivers, especially environmental concern, the following hypothesis can be developed.

**Hypothesis 13 (H13).** *Biospheric values have a significant and positive effect on environmental concern.*

2.4.2. Altruistic Values

Altruistic values are related to the concern for the welfare of others. They are based on understanding morals and show that a person intends to focus on others, rather than him/herself in making judgments regarding the environment and increasing the benefits to the general public, other people, and other living species [45,96]. According to Verma et al. [61], altruistic values have a significant and positive influence on environmental concern. Furthermore, Sánchez et al. [52] and Nguyen et al. [103] stated that altruistic values enable consumers to reduce pollution and buy green energy products by increasing their environmental protection attitudes. Individuals only make personal sacrifices to protect the environment when these actions help other humans. Altruistic behaviors, especially those relating to the sacrifice of one's interests for others' welfare, are a function of personal norms [104]. Altruistic values influence environmental attitudes, subjective norms, and perceived behavioral control [54]. Supposing that environmental concern is based on biospheric values, an individual will take action based on moral principles, including other species and nature, as goals [61]. Based on one of the RL drivers, namely environmental concern, the following hypothesis was formulated:

**Hypothesis 14 (H14).** *Altruistic values significantly and positively affect environmental concern.*

2.4.3. Egoistic Values

Egoistic values emphasize the maximization of individual benefits. They focus on how individuals value themselves in relation to other people and the environment and concentrate on their own welfare, such as strength and achievement, promoting short-term desires and long-term interests [45,96]. Egoistic values are related to the improvement of one's self or attributes. Therefore, people who show a strong egoistic orientation deliberately analyze certain actions in terms of the costs and benefits for themselves [61]. For instance, when the perceived benefits exceed the costs, green hotels are chosen and vice versa. Egoistic values affect environmental concern [61]. Therefore, the following hypothesis was formulated.

**Hypothesis 15 (H15).** *Egoistic values significantly and positively affect the environmental concern.*

**3. Methods**

*3.1. Sampling and Data Collection*

The study used a purposive sampling technique. Purposive sampling is a non-random sampling technique that determines the sample based on criteria that follow the research objectives so that the technique is expected to be able to answer the research problems [105]. The criteria used in this study are men and women aged 17 years and over who are smartphone consumers/users. This minimum age was adopted because consumers in this range can decide to participate in e-waste collection programs by bringing their used smartphones to collection centers. The data collection method used by the researchers was a questionnaire. A questionnaire consists of several written questions that are used to obtain information from the respondent in the sense of reports about his/her personality or things that he/she knows. Questionnaires can be in the form of closed or open questions/statements [106]. The questionnaire used in this study was a closed questionnaire. The distribution of the research questionnaire was carried out online through the Google form application from August to September 2020. Before the formal data collection, two trials were conducted. The first one tested the validity and reliability of the questionnaire

on 15 smartphone users. Comments and suggestions from these respondents were used to improve the questionnaire, specifically its ease of understanding. After the first revision, a second trial was carried out on 50 other users to evaluate its reliability. Finally, the questionnaire for data collection was created.

This study used Structural Equation Modeling (SEM). A total of 324 questionnaires were filled and processed to meet the recommended requirements. When the sample size is less than 100, almost any SEM type may not be sustainable, unless very simple models are evaluated [107]. Table 1 shows the sample distributions of the respondents.

**Table 1.** Sample distributions (n = 324).

| Variable | Category | Frequency | Percentage |
|---|---|---|---|
| Gender | Male | 139 | 42.90 |
| | Female | 186 | 57.10 |
| Ages | 17–25 years | 67 | 20.37 |
| | 26–35 years | 121 | 37.35 |
| | 36–45 years | 99 | 30.86 |
| | 46–55 years | 32 | 9.88 |
| | 56–65 year | 3 | 0.93 |
| | 66 years or older | 2 | 0.62 |
| Marital Status | Single | 210 | 64.81 |
| | Married | 110 | 33.95 |
| | Divorced/Widowed | 4 | 1.23 |
| Monthly Income (USD) | ≥1.000 | 33 | 10.19 |
| | 666.67–999.99 | 40 | 12.35 |
| | 333.33–666.66 | 104 | 32.10 |
| | 66.67–333.32 | 103 | 31.79 |
| | <66.67 | 44 | 13.58 |
| Education Degree | Doctoral degree | 12 | 3,70 |
| | Master's degree | 62 | 19.14 |
| | Bachelor's degree | 194 | 59.88 |
| | Diploma | 16 | 4.94 |
| | Senior high school | 35 | 10.80 |
| | Junior high school | 3 | 0.93 |
| | Primary school | 2 | 0.62 |

From the descriptive statistics shown in Table 1, the majority of respondents were female (57.10%), aged 26–35 years (37.35%), single (64.81%), and had a bachelor's degree (59.88%), with monthly income ranging from 66.67–333.32 USD to 333.33–666.66 USD (31.79% and 32.10%, respectively).

### 3.2. Measurement

The measurement variables shown in Table 2 were considered for each variable used in this study and were either selected or modified from previous studies. A total of eleven variables were used, including personal attitudes (6 indicators), subjective norms (4 indicators), perceived behavioral control (7 indicators), environmental concern (7 indicators), government drivers (4 indicators), economic drivers (3 indicators), facility accessibility (6 indicators), biospheric values (4 indicators), altruistic values (6 indicators), egoistic values (4 indicators), and consumer intention to participate in e-waste collection programs (5 indicators). The process of decreasing each variable/construct's indicators is based on the results of literature reviews from previous studies. For the biospheric values variable, there were three research results, namely those of Ghazali et al. [45], Verma et al. [61], and Shin et al. [96]. These scholars identified the indicators of the biospheric values variable as preventing pollution, respect for the earth, unity with nature, and protecting the environment. The results of decreasing the indicators of all the variables and references referred to can be seen in Appendix A. The Likert scale was used to measure the research variables. It is designed to test how strongly subjects agree or disagree with statements on a point scale, ranging from "strongly disagree" to "strongly agree" [108–110]. The questionnaire used a five-point Likert scale, ranging from 1 (strongly disagree/very not important/very unlikely) to 5 (strongly agree/very important/very likely). This scale asks respondents

to indicate how much they disagree or agree with something or whether they consider something to be inessential or essential, impossible or possible.

**Table 2.** Description of the measurement items in our questionnaire.

| Variables | Indicators | Mode | Median | Loading Factor | Cronbach's Alpha | AVE | CR |
|---|---|---|---|---|---|---|---|
| Biospheric Values (BV) | BV1 | 5 | 5 | 0.856 | 0.852 | 0.693 | 0.900 |
| | BV2 | 5 | 5 | 0.847 | | | |
| | BV3 | 5 | 5 | 0.847 | | | |
| | BV4 | 5 | 5 | 0.777 | | | |
| Altruistic Values (AV) | AV1 | 4 | 4 | 0.695 | 0.831 | 0.539 | 0.875 |
| | AV2 | 5 | 5 | 0.717 | | | |
| | AV3 | 5 | 5 | 0.809 | | | |
| | AV4 | 5 | 5 | 0.746 | | | |
| | AV5 | 4 | 4 | 0.761 | | | |
| | AV6 | 5 | 5 | 0.669 | | | |
| Egoistic Values (EV) | EV1 | 4 | 4 | 0.840 | 0.791 | 0.613 | 0.873 |
| | EV2 | 4 | 4 | 0.840 | | | |
| | EV3 | 4 | 4 | 0.609 | | | |
| | EV4 | 4 | 4 | 0.818 | | | |
| Environmental Concern (EC) | EC1 | 4 | 4 | 0.663 | 0.832 | 0.496 | 0.873 |
| | EC2 | 4 | 4 | 0.684 | | | |
| | EC3 | 4 | 4 | 0.802 | | | |
| | EC4 | 4 | 4 | 0.684 | | | |
| | EC5 | 5 | 5 | 0.750 | | | |
| | EC6 | 4 | 4 | 0.679 | | | |
| | EC7 | 4 | 4 | 0.655 | | | |
| Personal Attitudes (PA) | PA1 | 4 | 4 | 0.832 | 0.923 | 0.726 | 0.941 |
| | PA2 | 4 | 4 | 0.864 | | | |
| | PA3 | 4 | 4 | 0.717 | | | |
| | PA4 | 4 | 4 | 0.900 | | | |
| | PA5 | 4 | 4 | 0.905 | | | |
| | PA6 | 4 | 4 | 0.880 | | | |
| Subjective Norms (SN) | SN1 | 3 | 4 | 0.837 | 0.917 | 0.801 | 0.941 |
| | SN2 | 3 | 3 | 0.944 | | | |
| | SN3 | 3 | 3 | 0.930 | | | |
| | SN4 | 3 | 3 | 0.864 | | | |
| Perceived Behavioral Control (PC) | PC4 | 4 | 3.5 | 0.859 | 0.803 | 0.718 | 0.884 |
| | PC5 | 4 | 4 | 0.799 | | | |
| | PC6 | 4 | 4 | 0.882 | | | |
| Facility Accessibility (FA) | FA1 | 4 | 4 | 0.849 | 0.927 | 0.732 | 0.943 |
| | FA2 | 4 | 4 | 0.858 | | | |
| | FA3 | 4 | 4 | 0.877 | | | |
| | FA4 | 4 | 4 | 0.849 | | | |
| | FA5 | 4 | 4 | 0.870 | | | |
| | FA6 | 4 | 4 | 0.831 | | | |
| Economic Drivers (ED) | ED1 | 4 | 4 | 0.687 | 0.762 | 0.674 | 0.860 |
| | ED2 | 4 | 4 | 0.921 | | | |
| | ED3 | 4 | 4 | 0.839 | | | |
| Government Drivers (GD) | GD1 | 4 | 4 | 0.899 | 0.844 | 0.765 | 0.907 |
| | GD2 | 4 | 4 | 0.938 | | | |
| | GD3 | 4 | 4 | 0.780 | | | |
| Consumer Intention to Participate in e-waste Collection Programs (CI) | CI1 | 4 | 4 | 0.866 | 0.921 | 0.763 | 0.941 |
| | CI2 | 4 | 4 | 0.923 | | | |
| | CI3 | 4 | 4 | 0.870 | | | |
| | CI4 | 4 | 4 | 0.901 | | | |
| | CI5 | 4 | 4 | 0.800 | | | |

The mode and median values of each indicator are shown in Table 2. The highest mode value in the biospheric variable is 5 for all indicators, showing that Indonesians have a significant concern for nature and the environment. The mode value for the subjective norm variable (3 for all indicators) is lower than that for the other variables. This is because e-waste collection programs involving bringing used smartphones to collection centers are still not common in Indonesia.

### 3.3. Tools for Analysis

This study uses Variance Based Structural Equation Modeling (VB-SEM), with the Partial Least Squares–Structural Equation Modeling (PLS-SEM) technique. SEM is divided into two sub-models, including the measurement model consisting of different variables, to form potential variables and structural models, which refer to the linear regression model consisting of several potential variables [111]. The PLS design is intended to overcome the limitations of regression analysis using the OLS (Ordinary Least Square) technique, because there are problems with the data characteristics such as a small data size, missing values, abnormal form of data distribution, and the presence of multicollinearity symptoms. PLS-SEM was used in this study, because the sample size was less than 500, and the data were not normally distributed [112,113].

The research included (1) modeling, estimating, and simultaneously testing the indicator coefficient, which also means checking the validity and reliability of the variables' indicators; (2) modeling, estimating, and simultaneously testing the path coefficients between the existing variables; (3) testing the goodness of fit; and (4) testing the hypothesized relationship between the variables. SEM verification can be divided into two steps, i.e., verifying the measurement model and evaluating the reliability and validity of the potential variables. The measurement model evaluation index also includes reliability and validity. Furthermore, the model's goodness of fit is measured with a fitting index model, such as the standardized root mean square residual (SRMR) and R-square. The PLS method is used to verify the structural model, while the standard path coefficients and their significance are simultaneously calculated. The standard path coefficient refers to the degree of correlation between two variables. Supposing that the path coefficient is significant, the hypothetical relationship is supported [114].

The analysis tool used is smartPLS. SmartPLS is a free software of the PLS-SEM software packages; therefore, this software is freely available to research communities worldwide [115]. With PLS, confirmatory factor analysis (CFA) was used to assess the validity, reliability, and the model's goodness of fit, followed by SEM to test the hypothesis [116].

## 4. Results
### 4.1. Preliminary Analysis

In the data collection in this study, the respondents who assessed the predictor and those who assessed the dependent variables were the same person (self-reported methods). Self-reported methods can cause biases in judgment (e.g., the common method bias) [55]. The common method bias/variance is the systematic error variance between variables measured by a function of the same method and/or source [117]. A general method bias usually occurs in research when data for the independent and dependent variables are obtained from the same person in the context of a measurement using the same context items and similar item characteristics [118]. This bias can increase or decrease the observed correlation between constructs [119]. To ensure that a common method bias did not appear in this study, testing was needed to check for its presence. The widely used technique for testing for the common method bias is the Harman single-factor test [117,120]. This test incorporates all variables into an exploratory factor analysis and examines the non-rotated factor solution to determine the number of factors needed to explain the variance of the variables. A common method bias occurs when the test results show that there is one factor that can explain most of the data variance. The results of testing all variables using factor

analysis showed that one factor that was formed could explain 28.284% of the variance. Thus, it can be concluded that there was no common method bias in this study [121].

### 4.2. Reliability and Validity Analysis

Reliability refers to the internal consistency of the observed variables. Cronbach's Alpha is used to show the value of reliability. A Cronbach's Alpha coefficient value of 0.7 is acceptable, though the higher the value, the higher the reliability [122]. Where the Cronbach's Alpha coefficient exceeds 0.8, the internal consistency is excellent. Supposing that it ranges between 0.7 and 0.8, the internal consistency is good. However, if it is less than 0.7, the internal consistency is poor [123]. From the survey data, the Cronbach's Alpha coefficient for biospheric values, altruistic values, egoistic values, environmental concern, personal attitudes, subjective norms, perceived behavioral control, facility accessibility, government drivers, economic drivers, and consumer intention to participate in e-waste collection programs were 0.852, 0.831, 0.791, 0.832, 0.923, 0.917, 0.803, 0.927, 0.844, 0.762, and 0.921, respectively. All of the variables had Cronbach's Alpha coefficient values exceeding 0.7. All the variables also had a composite reliability (CR) value above 0.8. Therefore, all scales and subscales in this study had a good internal consistency and high reliability.

Validity analysis tests whether the observed variables can measure the latent variables. When the observed variable loading factor value is higher than 0.5, the latent variable has a good validity [122]. There are five indicators to be discarded from the processing results because they are below 0.5, namely, PC1, PC2, PC3, PC7, and GD4. Table 2 shows the complete results (with a significant probability value $p < 0.001$). The convergent validity is measured from the average variance extracted (AVE) value. Table 2 shows that the AVE value is greater than the threshold of 0.5 [122], and there is one variable whose value is close to 5 (0.496). Based on the loading factor and AVE values, the convergent validity is confirmed.

The discriminant validity is measured from the square root of the AVE in each latent variable. If this value of the same latent variable has a more significant correlation value than the other latent variables, the discriminant validity is well established [115]. Table 3 shows that the AVE's square root value on the same variable has a greater correlation value than the other variables, proving that the discriminant validity is well established.

**Table 3.** Discriminant validity test results.

| Variables | Item Indicators | Correlations | | | | | | | | | | |
|---|---|---|---|---|---|---|---|---|---|---|---|---|
| | | AV | BV | CI | ED | EV | EC | FA | GD | PC | PA | SN |
| AV | 6 | 0.734 | | | | | | | | | | |
| BV | 4 | 0.629 | 0.832 | | | | | | | | | |
| CI | 5 | 0.151 | 0.192 | 0.873 | | | | | | | | |
| ED | 3 | 0.055 | 0.099 | 0.111 | 0.821 | | | | | | | |
| EV | 4 | 0.318 | 0.227 | 0.152 | 0.233 | 0.783 | | | | | | |
| EC | 7 | 0.529 | 0.572 | 0.465 | 0.088 | 0.331 | 0.704 | | | | | |
| FA | 6 | 0.215 | 0.316 | 0.516 | 0.118 | 0.220 | 0.540 | 0.856 | | | | |
| GD | 3 | 0.211 | 0.295 | 0.634 | 0.096 | 0.202 | 0.414 | 0.510 | 0.875 | | | |
| PC | 3 | 0.141 | 0.199 | 0.514 | 0.147 | 0.194 | 0.401 | 0.418 | 0.566 | 0.847 | | |
| PA | 6 | 0.353 | 0.425 | 0.570 | 0.100 | 0.330 | 0.686 | 0.525 | 0.564 | 0.529 | 0.852 | |
| SN | 4 | 0.273 | 0.219 | 0.528 | 0.129 | 0.325 | 0.483 | 0.428 | 0.553 | 0.591 | 0.654 | 0.895 |

All CFA results indicate that the measurement model is convergent and acceptably discriminant. The hypothesized measurement model is reliable and examines the structural relationships between constructs/variables. The fit index was obtained from SRMR and R-square. A model has a good fit when the SRMR is less than 0.08 [113]. The R-square cut-offs must be 0.67, 0.33, and 0.19 to be substantial, moderate, and weak, respectively [113,124]. The SRMR value and R-square in this study were 0.063 and 0.506, respectively. The SRMR

value exceeded the general acceptance level, meaning that the measurement model aligns with the data. The R-square here would be considered to be of a moderate strength or effect.

### 4.3. Testing of the Structural Equation Model

SEM was conducted by SmartPLS to evaluate the conceptual model hypothesized from the study, as shown in Figure 2. Table 4 shows the SEM processing results, while the complete SEM processing results are shown in Appendix B. The SEM test results show the effect of the statement items in the questionnaire (indicators) on the latent variables. The indicator is said to have a positive and significant effect when the T-statistic value is greater than 1.96 [122]. From the figure in Appendix B, all indicators have a T-value greater than 1.96, so all indicators have a positive and significant effect on the latent variables. An indicator with a greater T-statistic value shows a more dominant influence than other statements/indicators on one latent variable. For example, the BV 3 indicator has the largest T-statistic value compared to the other indicators of the biospheric values variable. This shows that unity with nature has a more dominant influence on biospheric values than other variables.

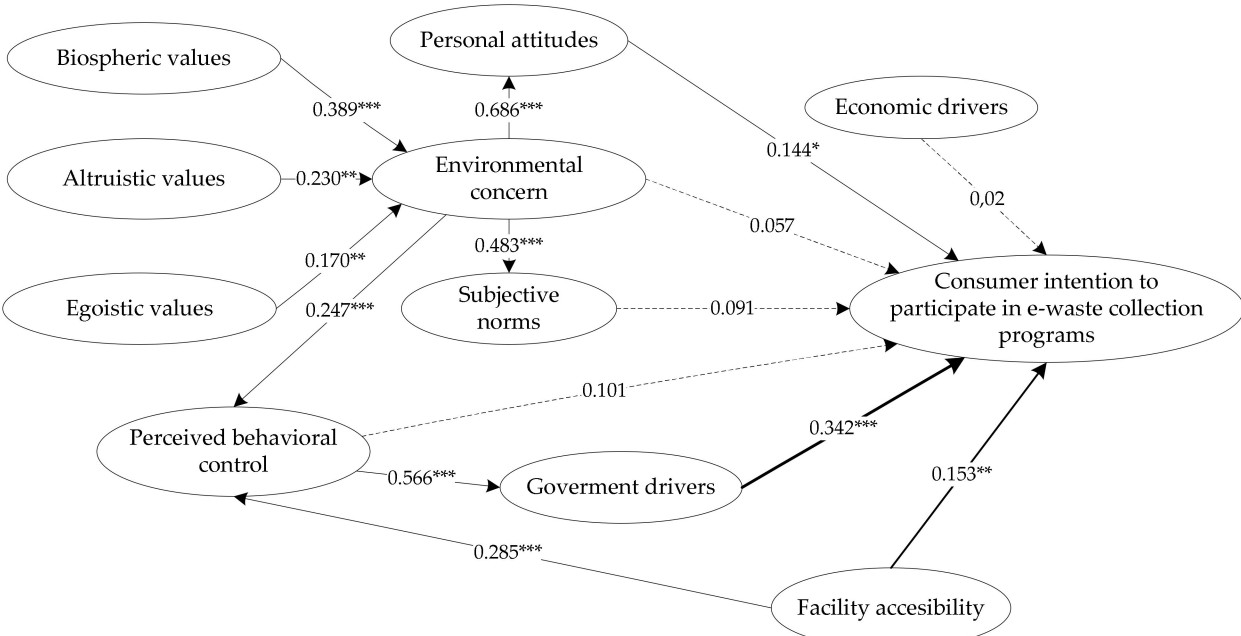

**Figure 2.** Structural equation model results. Notes: $p$ = significant probability, where * $p < 0.05$; ** $p < 0.01$; *** $p < 0.001$.

The structural model results and the standard path coefficients show a positive influence on the structural model constructs. In total, eleven of the fifteen hypotheses were supported. The effects of personal attitudes, subjective norms, perceived behavioral control, environmental concern, government drivers, economic drivers, and facility accessibility on consumer intention to participate in e-waste collection programs were (H1: $\beta_1 = 0.144$, $p = 0.032$), (H2: $\beta_2 = 0.091$, $p = 0.101$), (H3: $\beta_3 = 0.101$, $p = 0.071$), (H4: $\beta_4 = 0.057$, $p = 0.316$), (H8: $\beta_8 = 0.342$, $p < 0.001$), (H10: $\beta_{10} = 0.015$, $p = 0.774$), and (H11: $\beta_{11} = 0.153$, $p = 0.03$), respectively. This means that H1, H8, and H11 are supported, while H2, H3, H4, and H10 are rejected. According to H5, H6, and H7, the estimated positive coefficient of environmental concern for personal attitudes, subjective norms, and perceived behavioral control has a significant positive effect, which is (H5: $\beta_5 = 0.686$, $p < 0.001$), (H6: $\beta_6 = 0.483$, $p < 0.001$), and (H7: $\beta_7 = 0.247$, $p < 0.001$), respectively. Therefore, H5, H6, and H7 are supported. The effects of biospheric values, altruistic values, and egoistic values on environmental concern are (H13: $\beta_{13} = 0.389$, $p < 0.001$), (H14: $\beta_{14} = 0.230$, $p = 0.001$), and (H15: $\beta_{15} = 0.170$, $p < 0.001$), respectively. These values have a significant positive effect on environmental concern; therefore, H13, H14, and H15 are supported. Perceived behavioral control (H9:

$\beta_9 = 0.566$, $p < 0.001$) has a significant positive effect on government drivers; hence, H9 is supported. Facility accessibility has a significant positive effect on perceived behavioral control (H12: $\beta_{12} = 0.285$, $p < 0.001$), thus supporting H12. Environmental concern does not have a significant effect on consumer intention to participate in e-waste collection programs, but it has a significant indirect effect through personal attitudes, subjective norms, and perceived behavioral control, which is equal to 0.215. Likewise, perceived behavioral control does not have a significant effect on consumer intention to participate in e-waste collection programs. It has a significant indirect effect through government drivers of 0.194. Table 5 shows the direct, indirect, and total effects of each variable on consumer intention to participate in e-waste collection programs.

**Table 4.** Path coefficients for the structural model.

| Hypothesis | Paths | Estimate | S. E | *t*-Stat | *p*-Values |
|---|---|---|---|---|---|
| H1 | Personal Attitudes -> Consumer Intentions | 0.144 | 0.067 | 2.152 | 0.032 |
| H2 | Subjective Norms -> Consumer Intentions | 0.091 | 0.055 | 1.643 | 0.101 |
| H3 | Perceived Behavioral Control -> Consumer Intentions | 0.101 | 0.056 | 1.807 | 0.071 |
| H4 | Environmental Concern -> Consumer Intentions | 0.057 | 0.057 | 1.004 | 0.316 |
| H5 | Environmental Concern -> Personal Attitudes | 0.686 | 0.031 | 22.300 | 0.000 |
| H6 | Environmental Concern -> Subjective Norms | 0.483 | 0.044 | 11.015 | 0.000 |
| H7 | Environmental Concern -> Perceived Behavioral Control | 0.247 | 0.054 | 4.538 | 0.000 |
| H8 | Government Drivers -> Consumer Intentions | 0.342 | 0.056 | 6.094 | 0.000 |
| H9 | Perceived Behavioral Control -> Government Drivers | 0.566 | 0.040 | 14.081 | 0.000 |
| H10 | Economic Drivers -> Consumer Intentions | 0.015 | 0.045 | 0.327 | 0.744 |
| H11 | Facility Accessibility -> Consumer Intentions | 0.153 | 0.051 | 2.970 | 0.003 |
| H12 | Facility Accessibility -> Perceived Behavioral Control | 0.285 | 0.064 | 4.485 | 0.000 |
| H13 | Biospheric Values -> Environmental Concern | 0.389 | 0.057 | 6.844 | 0.000 |
| H14 | Altruistic Values -> Environmental Concern | 0.230 | 0.070 | 3.270 | 0.001 |
| H15 | Egoistic Values -> Environmental Concern | 0.170 | 0.050 | 3.385 | 0.001 |

Notes: Cut-off value: *t*-stat $\geq 1.96$, two-tiled sig 5% [111].

**Table 5.** The direct, indirect, and total effects of each variable on the consumer intention to participate in e-waste collection programs.

| Variables | Direct Effect | Indirect Effect | Total Effect | *p*-Value |
|---|---|---|---|---|
| Personal Attitudes | 0.144 | 0 | 0.144 | 0.032 |
| Subjective Norms | 0.091 | 0 | 0.091 | 0.101 |
| Perceived Behavioral Control | 0.101 | 0.194 | 0.295 | 0.000 |
| Environmental Concern | 0.057 | 0.215 | 0.272 | 0.000 |
| Government Drivers | 0.342 | 0 | 0.342 | 0.000 |
| Economic Drivers | 0.015 | 0 | 0.015 | 0.744 |
| Facility Accessibility | 0.153 | 0.084 | 0.237 | 0.000 |
| Biospheric Values | 0 | 0.106 | 0.106 | 0.000 |
| Altruistic Values | 0 | 0.063 | 0.063 | 0.005 |
| Egoistic Values | 0 | 0.046 | 0.046 | 0.005 |

## 5. Discussion

This study examined the framework of an extended TPB model, hereinafter referred to as the extended TPB. The values of the belief norm theory and environmental concern are added to the model as antecedents of the TPB variable. Facility accessibility is added as an antecedent of perceived behavioral control of consumer intention to participate in e-waste collection programs. This study also integrated TPB with reverse logistic drivers, including government drivers, economic drivers, and environmental concerns. The aim was to investigate the behavior of consumers over the age of 17 with respect to consumer intention to participate in e-waste collection programs by bringing their used smartphones to collection centers. The results showed that consumer intention to participate in e-waste

collection programs could be predicted by personal attitudes, subjective norms, perceived behavioral control, environmental concern, economic drivers, government drivers, and facility accessibility.

Government drivers, facility accessibility, and personal attitudes have a significant direct effect on consumer intention to participate in e-waste collection programs. Government drivers have the most significant influence on consumer intentions. This means that they are the strongest predictor of consumer intentions, followed by facility accessibility and personal attitudes. Perceived behavioral control does not have a significant direct effect on consumer intentions. It has a significant indirect effect on government drivers, which mediates between perceived behavioral control and consumer intentions. Likewise, environmental concern does not have a significant direct positive effect on consumer intentions. It indirectly affects consumer intentions through TPB variables, which include personal attitudes, subjective norms, and perceived behavioral control. Furthermore, environmental concern is significantly influenced by biospheric values, altruistic values, and egoistic values. This shows that it is necessary to integrate RL drivers and VBNT in the TPB framework, because the variables reinforce each other. The results of this study are in line with those of Nguyen and Johnson [125], who indicated that the determinants of pro-environmental behavior are complex, and an integrative model is therefore needed.

Government drivers constitute the most significant factor and illustrate the regulations regarding the e-waste collection programs by bringing their used smartphones to the collection centers. Currently, Indonesia is about to apply these rules; hence, the question in the questionnaire used the words, "if it is actually applied". The results show that this factor has the highest influence on consumer intention to participate in e-waste collection programs. This means that if the Indonesian government makes special rules to support e-waste collection programs, these rules will significantly encourage consumers to participate in the program. Therefore, consumers need encouragement in the form of government regulations to participate in e-waste collection programs. This is in line with Yuan [124], who established that government regulation is one of the critical factors in waste management success. Compliance with regulations is the most vital element in promoting the use of modes of construction and demolition that are compatible with recycling [25]. Legal and regulatory policies are the main instruments for the implementation of e-waste management. However, these policies are not well implemented by the electronics industry [90]. According to Zhang et al. [36], government stimulus has a positive and significant effect on waste sorting behavior.

Facility accessibility has a significant effect, either directly on consumer intention to participate in e-waste collection programs or indirectly through PCB. This shows that people need facility accessibility to participate in e-waste collection programs. Facility accessibility includes short distances and travel times, directions to the location of the used smartphones collection centers, road conditions, ease of use of collection centers, and collection centers capacity. In general, facility accessibility support from the government has improved consumer intentions. This is in line with the findings presented in Zhang et al. [36], in which it was concluded that facility accessibility significantly influences waste sorting behavior. According to Zhang et al. [48], facility accessibility affects individual intentions to recycle waste. Ease of access in the form of doorstepping increases individual intentions to engage in waste recycling behavior. A waste collection service, involving the picking up of waste from a site in Shanghai, China, increased the average recycling growth by 12.5% [126]. In general, shorter distances mean easier access to sorting facilities, thus encouraging personal behavior. Perceived distance is a more important factor influencing recycling behavior [93,94]. Determining the location of waste facilities is one of the steps in waste management planning [127,128].

The third-order factor that significantly influences consumer intention to participate in e-waste collection programs is personal attitudes. Compared to government drivers and facility accessibility, personal attitudes are intrinsic factors that reflect individual subjective desires. They deal with self-expectations and individual attitudes regarding

specific behaviors that stem from norms and values related to behavior. These results indicate that individual attitudes towards the environment influence consumer intention to participate in e-waste collection programs. Therefore, it is necessary to include the VBNT personal norm as a TPB attitude variable. These results are in line with the TPB presented by Ajzen [68], as a reference in modeling behavior. Furthermore, the results are also in line with those presented in previous studies, which showed a positive and significant influence on personal attitudes towards waste sorting behavior intentions and reducing car travel [36,49]. This is also reinforced by previous studies that showed that attitudes have a positive and significant effect on the intention to visit green hotels [61], the intention to purchase eco-friendly packaged products [64], the intention to recycle [21,23,24], ecolabel product usage [86], purchasing reusable shopping bags [28], green behavior [32], green purchasing [29], the behavioral intention to buy green products [30,63], and the intention to purchase green products [62]. Personal norms significantly influence behavioral intentions [41], reduce intentions to consume clothing [38,41], and increase the intention to engage in green purchasing behavior [47]. Some studies stated that attitudes do not affect the intention to use cars for traveling to campus, recycling intentions, and intentions to purchase green products [22,50,65]. According to Macovei [72], attitudes harm intentions to conserve energy.

Environmental concern does not directly affect influence consumer intention to participate in e-waste collection programs, but it indirectly affects consumer intention to participate in e-waste collection programs through the three TPB variables, including personal attitudes, perceived behavioral control, and subjective norms. It has a significant positive effect on personal attitudes, subjective norms, and perceived behavioral control. Environmental concern has been studied in many different contexts, including environmental worldviews, ecological or environmental values [129], and motivating people to participate in environmentally friendly activities and consider themselves as saviors of the environment [130]. This is a stage in environmental literacy that helps determine people's values and attitudes regarding the environment and uses them to guide behavior [131]. This study's results are in line with those presented in Setyawan et al. [65], and Paul et al. [63], which also showed that environmental concern has an indirect positive and significant effect on consumer intentions but does not directly influence the intention to buy green products. Similarly, Verma et al. [61] and Dursun et al. [43] established that the effect of environmental concern is stronger on attitudes than directly on intentions to visit green hotels. The two studies also concluded that environmental concern has a strong effect on personal norms. According to Paul et al. [63], the effect of environmental concern on personal attitudes, subjective norms, and perceived behavioral control is greater than the direct effect on purchase intentions.

Perceived behavioral control does not directly affect influence consumer intention to participate in e-waste collection programs but has a significant and indirect positive effect through government drivers. Primarily, perceived behavioral control is the ease or difficulty that a person feels in carrying out certain behaviors [62]. Currently, many people do not know about e-waste collection programs. A small number of cities have provided collection center facilities; hence, the community has not assessed the ease or difficulty they face, which necessitates regulations. This finding is in line with Arifani and Haryanto [28], who established that the influence of perceived behavioral control on the intention to buy reusable shopping bags is not strong. Similarly, Budijati [56] showed that the direct effect of perceived government regulations/support on environmental behavior is insignificant and not strong. Furthermore, this study is supported by Nadlifatin et al. [66] and Persada et al. [67], who showed that the indirect effect of perceived government regulations/support on environmental behavior is vital and significant. However, several studies show that perceived behavioral control has a positive and significant effect on intentions, such as the intention to recycle [21,23], use cars for traveling to campus [50], reduce car travel [49], engage in green behavior [32], conserve energy [72], and purchase green products [29,30,62,63,65].

Subjective norms do not affect influence consumer intention to participate in e-waste collection programs, because only a small number of cities have provided e-waste collection centers. Many people do not know and are not familiar with the program. Due to their ignorance, families, colleagues, friends, and important people cannot remind, encourage, or invite each other to participate in the program. This is in line with previous studies, which concluded that subjective norms do not affect willingness to pay to reduce noise pollution in land transportation [52], the intention to reduce car travel, energy conservation behavior, and the intention to purchase green products [49,51,63]. However, several studies provided different results. Subjective norms have a positive and significant effect on intention, including the intention to use ecolabel products, reduce the use of cars to travel to campus, purchase reusable shopping bags, purchase green products, recycle, and sort waste [23,24,29–31,36,50,62,65,86].

Economic drivers also do not affect influence consumer intention to participate in e-waste collection programs. Economic drivers describe offers of financial benefits to consumers if they participate in waste collection programs. Therefore, incentives do not influence consumer intentions to participate in e-waste collection programs. This assertion is reinforced by the fact that economic drivers received mode values of 3 on a scale of 5, showing that the respondents were willing to bring their used smartphones to collection centers without expecting incentives. There is no incentive for consumers to bring their used smartphones to the collection centers. Previous studies have shown that economic incentives are quite successful in influencing consumer environmental behavior, including the adoption of electric vehicles, saving energy, and waste separation [132–134]. However, this study is in line with Wang et al. [135] from Shenzhen, who stated that economic incentives are the least important factor.

Biospheric, altruistic, and egoistic values have a positive and significant effect on environmental concern. According to Verma et al. [61], these three values have a significant positive effect on environmental concern. Furthermore, Lind et al. [100] showed that biospheric and egoistic values positively and negatively affect the new environmental paradigm. Biospheric values have a positive and significant effect on environmental concern. Biospheric values include preventing pollution, respecting the earth, unity with nature, and protecting the environment. This is in line with Verma et al. [61], Lind et al. [100], and Han et al. [46], who stated that biospheric values significantly affect environmental concern. Furthermore, Hiratsuka et al. [97] and Ghazali et al. [45] used the VBNT model and established that biospheric values positively and significantly affect awareness of environmental consequences. Altruistic values positively and significantly affect environmental concern and include equality, a peaceful world, social justice, help, loyalty, and respecting parents or the elderly. Similarly, Verma et al. [61] stated that altruistic values have a significant positive effect on environmental concern, though other studies show that they have a significant positive effect on the awareness of environmental consequences and a negative effect on personal norms [45,102]. Conversely, the result of Han et al. [46] showed that altruistic values do not significantly affect ecological worldview. Egoistic values, which include influence, social strength, wealth, and authority, positively and significantly affect environmental concern. This is in line with Verma et al. [61] and strengthened by Sánchez et al. [52] and Kim and Seock [136], who showed that egoistic values have a positive effect on attitudes and personal norms, while Lind et al. [100], Han et al. [46], Hiratsuka et al. [97], and Lauper et al. [102] showed a negative effect. The results of Ghazali et al. [45] showed that egoistic values do not affect awareness of environmental consequences.

This study provides several implications that can aid in the development of the right strategy for promoting influence consumer intention to participate in e-waste collection programs by bringing their used smartphones to collection centers. Government drivers, facility accessibility, and personal attitudes have a significant effect on influencing consumer intention to participate in e-waste collection programs. Government drivers have the strongest influence, followed by facility accessibility and, finally, personal attitudes.

Because government drivers constitute the factor that has the most substantial influence on consumer intention to participate in e-waste collection programs, it is time for Indonesia to introduce regulations for the handling of electronic waste. Some countries have already established regulations regarding the re-management of electronic products that are no longer used by consumers, such as the WEEE (Waste Electrical and Electronic Equipment) Directive 2002, which regulates the collection and management of environmentally friendly electronic equipment for countries in Europe; Royal Decree 208/2005; the regulations for WEEE collection and handling in Spain [137]; the EPA-HQ-RCRA-2004-0012 regulations, namely, the Hazardous Waste Management System, Modification of the Hazardous Waste Program, Cathode Ray Tubes, and Final Rule in the United States [138]; the Ordinance on the Return, Taking back, and Disposal of Electrical and Electronic Equipment (ORDEE) applied in Switzerland; the Environmental policy agreement on the mandatory take-back of waste from electrical and electronic equipment, which is implemented in Belgium [139]; the San Luis Obispo (SLO) system, which mandates a retail take-back program for household hazardous waste and materials (HWWM) in the state of California [140]; and E-waste Management and Handling Rules in India [141]. Electronic waste handling in China is regulated by the Regulations on the Recovery Processing of Waste from Electrical and Electronic Products [142]. Malaysia has the Environmental Quality (Scheduled Wastes) Regulations 2005 [143]. Meanwhile, Japan has the Law for the Promotion of the Effective Utilization of Resources (LPUR) and the Law for Recycling Specified Kinds of Home Appliances (LRHA) [144].

Sweden has one of the best waste management systems in the world. It can be considered a reference country, as it is very concerned with the waste problem. Sweden is also the country with the highest recycling rate in the world, accounting for 99% of its waste [145]. In Sweden, there is a government policy that stipulates that a waste recycling center must be available within 300 m of any settlement. This policy aims to orient community behavior toward disposing and treating waste properly [146]. E-waste collection in Sweden shows a collection rate of over 60% [147]. This Swedish policy can become a reference for waste management in Indonesia, because the accessibility of public facilities is a determining factor for influence consumer intention to participate in e-waste collection programs. The government must provide e-waste collection center facilities at an affordable distance for the community to recycle their e-waste. Considering Indonesia's more expansive territory in comparison with that of Sweden, the distance does not have to be as close as that employed in the Swedish policy, but it must still be affordable for people to access. The regulations in Malaysia stipulate that people are prohibited from throwing electronic waste into landfills; electronic waste must be recycled and recovered in a designated and licensed place. Through the Ministry of Environment (DOE), Malaysia has issued permits for 18 full recovery facilities and 128 partial recovery facilities to convert various types of electronic waste into source materials [143]. To ensure recycling activities do not endanger the environment, Indonesia can refer to Malaysia's electronic waste management system; only licensed recyclers can recycle. Moreover, the number of recycling bins should be set up as needed, taking into account the volume of waste. This should be conducted in accordance with Malaysian regulations. In Malaysia, many full recyclers cannot operate at full capacity due to a lack of e-waste supply. In campaigning for the e-waste collection program, the government can use environmental issues that touch the community or consumers and collaborate with non-governmental organizations. When Indonesia succeeds in managing its e-waste properly, it is hoped that the environment will be clean, that there will be no soil and environmental pollution, and that there will be no flooding due to clogged garbage.

## 6. Conclusions

The extended TPB variable influences consumer intention to participate in e-waste collection programs by bringing their used smartphones to collection centers. The results showed that government drivers constitute the most influential factor in the extended TPB model, followed by facility accessibility and, finally, personal attitudes. Therefore, government drivers greatly influence consumer intention to participate in e-waste collection programs in Indonesia. Furthermore, facility accessibility is the second most influential factor, after government drivers. The third most influential factor that significantly influences consumer intentions is personal attitudes. Environment concern positively influences consumer intentions through the Theory of Planned Behavior variables. Meanwhile, perceived behavioral control positively influences consumer intentions through government drivers. The results show the need for integration, because the variables reinforce each other. However, neither economic drivers nor subjective norms significantly influence consumer intentions. This finding distinguishes Indonesia from other countries, especially developed countries, because e-waste collection programs have not become part of the culture in Indonesia; therefore, Indonesia needs regulations, as the most influential variable, to regulate the implementation of such a program. Based on this study's findings, the Indonesian government is expected to draft e-waste management regulations that will oblige consumers to participate in e-waste collection programs and set up collection centers that are easy and affordable to access. To orient personal attitudes in a sustainable direction requires campaign enforcement. The government can take advantage of environmental issues that affect the community in campaigning for e-waste collection programs.

This study has several limitations that need to be resolved in future research. For instance, it only focused on smartphones. Thus, future research should extend the findings of this paper to other products with a larger volume, such as refrigerators, washing machines, televisions, etc. The greater volume would also help to determine the effectiveness of the collection process. Moreover, this research added only RL drivers and the values of a new environmental paradigm to extend the TPB, and other variables could be included in the future. This may include knowledge, past experiences, inconvenience, openness to change, awareness, responsibility, environmental awareness and knowledge, and social and moral norms [22,38,45,97,136,148]. These factors can be considered variables that may be added to the TPB in order to analyze consumer intention relating to the e-waste collection programs more comprehensively.

**Author Contributions:** Conceptualization, D.P.S., N.A.M. and A.M.S.A.; methodology, D.P.S., N.A.M. and A.M.S.A.; software, D.P.S.; validation and analysis, D.P.S., N.A.M. and A.M.S.A.; data curation, D.P.S.; writing—original draft preparation, D.P.S.; writing—review and editing, N.A.M. and A.M.S.A.; visualization, D.P.S.; supervision, N.A.M. and A.M.S.A. All authors have read and agreed to the published version of the manuscript.

**Funding:** This research was funded by the BUDI DN Grant from the Lembaga Pengelola Dana Penelitian (LPDP), Ministry of Finance, and Ministry of Education and Culture, Republic of Indonesia, grant number KET-193/LPDP.4/2020.

**Institutional Review Board Statement:** Not applicable.

**Informed Consent Statement:** Not applicable.

**Data Availability Statement:** The data used to support the findings of this study are available from the corresponding author by request.

**Acknowledgments:** We would like to thank the survey respondents for taking the time to answer the questionnaire and the reviewers of this paper for their valuable comments.

**Conflicts of Interest:** The authors declare no conflict of interest.

## Appendix A

| Variables | The Measurement Items | Sources |
|---|---|---|
| Biospheric Values (BV) | Preventing Pollution (BV1) | [45,61,96] |
| | Respect for the Earth (BV2) | |
| | Unity with Nature (BV3) | |
| | Protecting the Environment (BV4) | |
| Altruistic Values (AV) | Equality (AV1) | [45,61,96] |
| | A Peaceful World (AV2) | |
| | Social Justice (AV3) | |
| | Helpful (AV4) | |
| | Loyalty (AV5) | |
| | Respect for Parents and Elders (AV6) | |
| Egoistic Values (EV) | Influence (EV1) | [45,61,96] |
| | Social Strength (EV2) | |
| | Wealth (EV3) | |
| | Authority (EV4) | |
| Environmental Concern (EC) | I am very concerned about the state of the world environment (EC1) | [61–63] |
| | I am willing to participate in e-waste collection programs by bringing my used smartphones to collection centers for protecting the environment (EC2) | |
| | Major social changes are needed to protect the natural environment (EC3) | |
| | Major political changes are needed to protect the natural environment (EC4) | |
| | Humans must maintain a balance with nature in order to survive (EC5) | |
| | Human disturbance of nature often has disastrous consequences (EC6) | |
| | Anti-pollution laws need to be more strongly enforced (EC7) | |
| Personal Attitudes (PA) | I think e-waste collection programs by bringing my used smartphones to collection centers is useful and profitable (PA1) | [36,62,63] |
| | I think participate in e-waste collection programs by bringing my used smartphones to collection centers is important for protecting the environment (PA2) | |
| | I think participating in e-waste collection programs by bringing my used smartphones to collection centers is useful for reducing natural resource shortages (PA3) | |
| | I think participating in e-waste collection programs by bringing my used smartphones to collection centers is a good idea (PA4) | |
| | I like the idea of participating in e-waste collection programs by bringing my used smartphones to collection centers (PA5) | |

| Variables | The Measurement Items | Sources |
|---|---|---|
| | I support participating in e-waste collection programs by bringing my used smartphones to collection centers (PA6) | |
| Subjective Norms (SN) | My family wants me to participate in e-waste collection programs by bringing my used smartphones to collection centers (SN1) | [36,62,63] |
| | My close friend thinks that I should participate in e-waste collection programs by bringing my used smartphones to collection centers (SN2) | |
| | My colleagues think that I should participate in e-waste collection programs by bringing my used smartphones to collection centers (SN3) | |
| | Most people who are important to me think that I should participate in e-waste collection programs by bringing my used smartphones to collection centers (SN4) | |
| Perceived Behavioral Control (PC) | I feel uncomfortable participating in e-waste collection programs by bringing my used smartphones to collection centers (PC1) | [36,62,63] |
| | The decision of whether I participate in e-waste collection programs or not is entirely up to me (PC2) | |
| | I do not know how to participate in e-waste collection programs (PC3) | |
| | I have the resources, time, will, and opportunity to participate in e-waste collection programs by bringing my used smartphones to collection centers (PC4) | |
| | I believe that if I want, I can participate in e-waste collection programs by bringing my used smartphones to collection centers (PC5) | |
| | I believe that I can participate in e-waste collection programs by bringing my used smartphones to collection centers (PC6) | |
| | I feel that participating in e-waste collection programs by bringing my used smartphones to collection centers is not entirely within our control (PC7) | |
| Facility Accessibility (FA) | Distance to affordable collection centers around the community where they live (FA1) | [36,93,95] |
| | There are instructions directing one to the location of collection centers (FA2) | |
| | Travel time to the location of collection centers (FA3) | |
| | Road conditions to collection centers (FA4) | |
| | Ease of use of collection centers (FA5) | |
| | The capacity for collection centers is sufficient (FA6) | |
| Economic Drivers (ED) | Consumers who participate in e-waste collection programs by bringing their used smartphones to collection centers need to be given cash compensation (incentives) (ED1) | [13,56] |

| Variables | The Measurement Items | Sources |
|---|---|---|
| | Consumers who participate in e-waste collection programs by bringing their used smartphones to collection centers need to be given replacements (incentives) as discounts in purchasing new products (ED2) | |
| | Consumers who participate in e-waste collection programs by bringing their used smartphones to collection centers need to be given replacements (incentives) as new products by adding certain costs (ED3) | |
| Government Drivers (GD) | If the government actually implements rules that urge consumers to bring their used smartphones to collection centers to be managed in order to prevent environmentally damaging disposal, you intend to participate in the collection program (GD1) | [19,56] |
| | If the government actually enforces a rule that encourages consumers to bring their used smartphones to collection centers to be managed in order to prevent environmentally damaging disposal, then you intend to participate in the collection program (GD2) | |
| | If the government actually enforces a rule requiring bring their used smartphones to collection centers to be managed in order to prevent environmentally damaging disposal, then you intend to participate in the collection program (GD3) | |
| | If the government really applies sanctions/fines for consumers who do not bring their used smartphones to collection centers to recycle their used smartphones, you intend to participate in the collection program (GD4) | |
| Consumer Intention to Participate in e-waste Collection Programs (CI) | I plan to participate in e-waste collection programs by bringing my used smartphones to collection centers (CI1) | [36,62,63] |
| | I am willing to participate in e-waste collection programs by bringing my used smartphones to collection centers (CI2) | |
| | I would consider participating in e-waste collection programs by bringing my used smartphones to collection centers (CI3) | |
| | I hope to participate in e-waste collection programs by bringing my used smartphones to collection centers (CI4) | |
| | I want to participate in e-waste collection programs by bringing my used smartphones to collection centers soon (CI5) | |

**Appendix B**

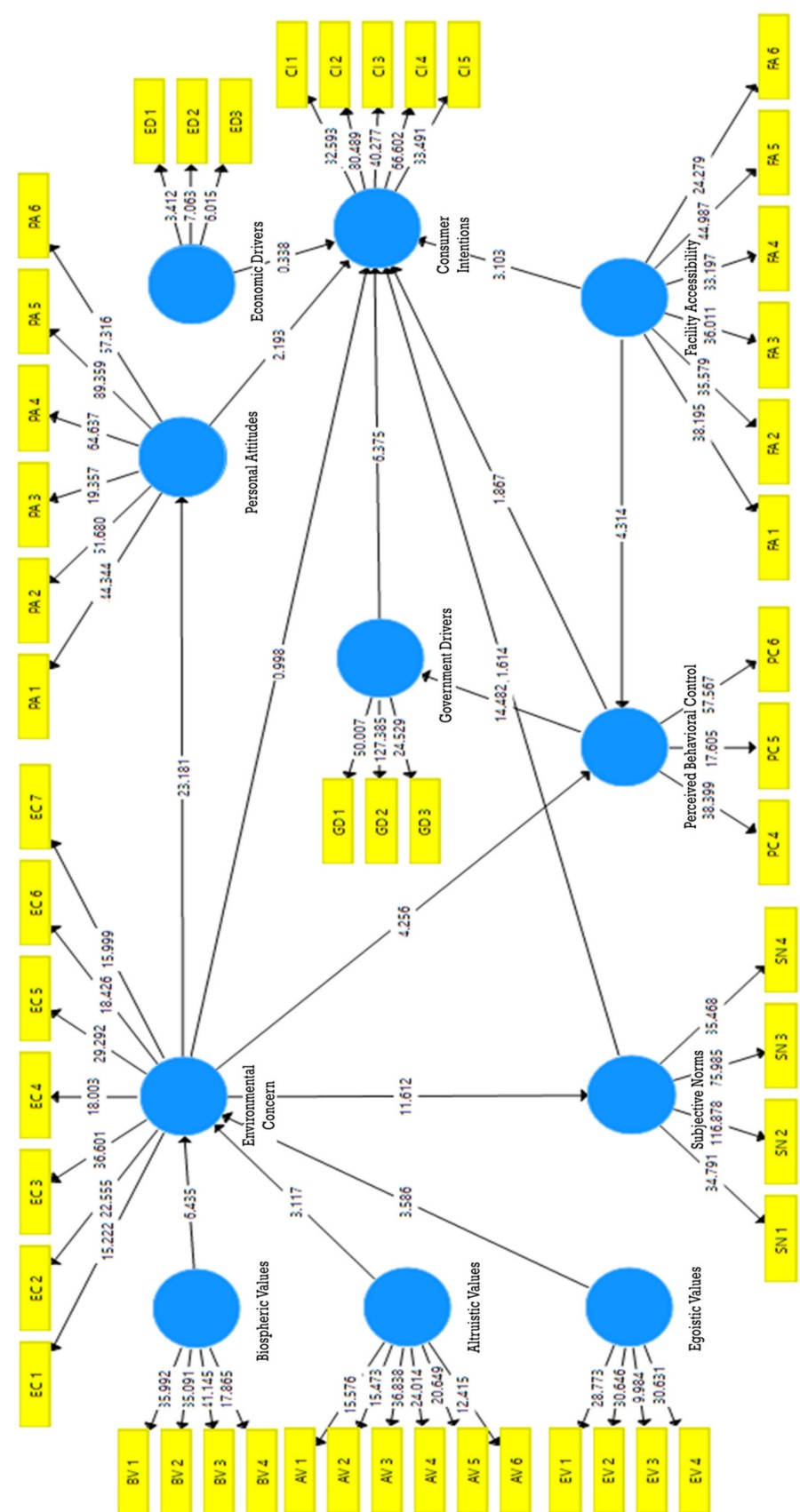

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
