# Peer review of "Consumer Intention to Participate in E-Waste Collection Programs: A Study of Smartphone Waste in Indonesia"

_sustainability, doi:10.3390/su13052759_

Round 1
Reviewer 1 Report
Overall this an interesting study that aims to improve understanding of consumers' behaviour such that progress towards a more resource-efficient approach can be made towards e-waste. There are several issues that I consider need to be addressed or amended as follows:
The title indicates the paper considers e-waste, but the focus here is on smartphones specifically. The title should be amended to reflect the focus.
The focus on promoting environmentally sustainable behaviour needs much greater emphasis and focus, given the theme of the special issue for which it has been submitted. How do the outcomes of this study help in to progress towards environmental sustainability? What difference could it make to consumers' behaviour and what would the impacts be in terms of environmental sustainability?
It is assumed at an early part of the manuscript that reverse logistics offer a solution to the collection of e-waste for reuse or recycling. This position needs to be be more fully supported and justified: what are the alternative approaches and why does RL merit inclusion in this study?
The setting out of hypotheses works well and makes clear the reasoning behind the hypotheses at hand. However, given that the TPB forms the "backbone" of the model as tested, it would be appropriate to set out the TPB factors at the start of this section of the methods and then add in other factors. Figure 1 works well as a representation of the extended TPB model, but the caption needs more detail: describe what is shown more fully and include comment that h1, h2 etc. indicate hypotheses - and cross-reference to the relevant section of the paper. Figure 1 indicates that the study address consumers' intention to collect e-waste - surely they are not collecting e-waste but taking it somewhere for subsequent reuse or recycling.
There is no statement regarding the desired respondent group. Was the intention to gather responses form a group that is representative of a specific population? The profile of the respondent group (Table 1) shows only the respondent group and not a broader population. The survey was intended to respondents over 17 years of age, but Table 1 indicates that some respondents were of school age; this needs to be commented upon or responses removed and statistics re-worked. For an international audience, the income values shown on table 1 will have little meaning. It would be instructive to include some context here e.g. convert income values to US dollars or Euros, or indicate maybe average earnings in Indonesia as a benchmark.
The methods indicate that Likert scale questions were used, and then average "scores" for responses are given. Is it legitimate to use average values for Likert scale questions? Some comments are needed in this regard: it is common practice to do this, but the practice has been challenged by some statisticians. Where average values are given (Table 2), a value of N should be given or standard error given in preference to standard deviation.
Table 2 is rather large and I would envisage difficulties formatting this so that it would fit onto a page of the journal. It may be worthwhile presenting this as supplementary information, with a smaller table presenting selected highlights or an overview/summary of this table in the main text. The bullet points in table 2 are superfluous.
Table 4 might not be necessary. The key outcomes of the study are, in essence, shown in figure 2 and table 4. Could the data shown in table 4 be added (as annotations) for figure 2? If this were done then the strength of interactions (as tested hypotheses) could be viewed on the same template as the model being elucidated. Annotations could easily incorporate both the regression weights and the associated P values (*/**/***). As currently presented, it is not clear that the numbers shown as annotations on figure 2 are the regression weights; this needs to be made clear in the caption for figure 2.
The discussion is generally coherent and effective. The themes explored in the last paragraph do, I believe, merit expansion. In particular the proposals for Indonesia need to be considered in the context of "received wisdom", notably some commentary on where and how the proposed approaches have been used elsewhere - did they work, why did they work and how does this inform their potential application in Indonesia?
The conclusion section is rather long - this repeats some of the outcomes of the study that are represented in fuller and better detail earlier in the paper. I would prefer a shorted conclusion with a more concise and purposeful statement of the key outcomes and applications of the study.
Author Response
Dear Reviewer #1
In this letter, we offer our responses to your comments on the manuscript submitted to Sustainability, for a Special Issue of Environmentally Sustainable Behavior: Theories, Empirical Evidence and Implications entitled 'An Extended of Theory Planned Behavior Model to Explain the Customer Intentions to Collect e-waste in Indonesia'
We thank you for your insightful and important comments and giving us the opportunity to expand on elements of the manuscript we had not addressed. We have provided point-by-point responses to your comments, which we present in table form. Our response is attached, please see the attachment. Hopefully our responses is as you expect.
We once again thank you for your constructive comments and recommendations on this manuscript.
Kind regards,
Diana Puspita Sari, Nur Aini Masruroh and Anna Maria Sri Asih

Reviewer 2 Report
The paper entitled “An Extended of Theory Planned Behavior Model to Explain the Customer Intentions to Collect E-waste in Indonesia" by Diana Puspita Sari, Nur Aini Masruroh and Anna Maria Sri Asih, appears to be incomplete in the analyzes, therefore it needs major revisions to be published in Sustainability. The most important points, which the authors should necessarily review in the paper, are clarified below. 1. The authors should clearly justify why, among the different models (eg Partial Least Squares Structural Equation Modeling PLS-SEM), they decided to use the “Covariance Based-Structural Equation Modeling (CB-SEM)”; above all, they should stress what are the limits of application of the model used. 2. The authors state "... the study used an intentional sampling technique ..." (in order to justify the use of only 324 samples). However, a CB-SEM model should only be used for normally distributed samples (Vinzi, et Al., Handbook of Partial Least Square). In these cases, in fact, a better alternative could be the PLS -SEM method (Wong, Marketing Bulletin, 2013, 24, Technical Note 1). The authors should clearly explain why they did not use this method. 3. Authors should append to the end of the paper Report the administration questionnaire. They should specify in detail and comment on the criteria by which the questionnaire was formed. 4. Authors should indicate how they related items of the questionnaire to variables/indicators of the statistical model. 5. Plots in Appendix A, should be made more readable (increase the contrast).Author Response
Dear Reviewer #2
In this letter, we offer our responses to your comments on the manuscript submitted to Sustainability, for a special issue of Environmentally Sustainable Behavior: Theories, Empirical Evidence and Implication entitled 'An Extended of Theory Planned Behavior Model to Explain the Customer Intentions to Collect e-waste in Indonesia'.
We thank you for your insightful and important comments and giving us the opportunity to expand on elements of the manuscript we had not addressed. We have provided point-by-point responses to your comments, which we present in table form. Our response is attached, please see the attachment. Hopefully our responses is as you expect.
We once again thank you for your constructive comments and recommendations on this manuscript.
Kind regards,
Diana Puspita Sari, Nur Aini Masruroh, Anna Maria Sri Asih

Round 2
Author Response
Dear Reviewer
Thank you for your constructive comments and recommendations on this manuscript.
Kind regards,
Diana Puspita Sari, Diana Puspita Sari and Anna Maria Sri Asih